# Comment on "Are soils overrated in hydrology?" by Gao et al. (2023)

**Ying Zhao[1,2], Mehdi Rahmati[3,4], Harry Vereecken[4], Dani Or[5,6]**

[1]Dongying Base of Integration between Industry and Education for High-quality Development of Modern Agriculture, Ludong University, Dongying 257509, China
[2]College of Resources and Environmental Engineering, Ludong University, Yantai 264000, China
[3]Department of Soil Science and Engineering, University of Maragheh, Iran
[4]Agrosphere Institute IBG-3, Forschungszentrum Jülich GmbH, Germany
[5]Department of Environmental Systems Science, ETH Zurich, Switzerland
[6]Department of Civil and Environmental Engineering, University of Nevada Reno, USA

**Correspondence:** Ying Zhao (yzhaosoils@gmail.com).

**Abstract.** This comment challenges Gao et al. (2023)'s perspective rejecting the role of soil processes in hydrology. We argue that the authors present a false dichotomy between soil-centric and ecosystem-centric views. These two views of hydrology are complementary and reflect on the inherent multiscale complexity of hydrology where soil processes dominate at certain scales but other processes may become important at catchment scale. We recognize the need for a new scale-aware framework that reconciles the interplay between soil processes at small scales with emergent behaviors driven by vegetation, topography and climate at large scales.

## 1 Introduction

The recent HESS Opinions paper by Gao et al. (2023) "Are soils overrated in hydrology?" offers a provocative perspective. While we agree with certain points raised in their piece, we welcome the opportunity to challenge sweeping and poorly substantiated assertions regarding the role of soil processes in hydrology. Our response is organized around 3 main points: (1) separation of ecosystem-centerd hydrology from soil-centered representation offers a false dichotomy; (2) we highlight the importance and limitations of soil properties across different scales; and (3) we argue for the need of a scale-aware theoretical framework to replace the current reliance of watershed hydrology on small-scale soil processes, a framework that interfaces naturally with soil physics where appropriate. We conclude by suggesting ways to reconcile the two perspectives.

## 2 A false dichotomy?

In his seminal work "a random walk on water", Koutsoyiannis laments the traditional dichotomy in science between determinism ("good") and randomness ("evil") and concludes that the "entire logic of contrasting determinism with randomness is just a false dichotomy" (Koutsoyiannis, 2010). Similarly, we argue that the division presented in Gao et al. (2023), contrasting soil-centered (microscale) and ecosystem-centered (macroscale) views of hydrology, represents a false dichotomy that hinders a deeper and nuanced understanding of hydrology as the study of water in nature. Hydrology's inherently multiscale character demands observational and theoretical approaches for describing processes at all scales, from water distribution within a single pore to the behavior of river networks at continental scales. Prioritizing one scale or process over another unnecessarily limits the scope of contemporary hydrology.

The assertions in Gao et al. (2023), such as "*the ecosystem, not the soil, determines the land-surface water balance and hydrological processes. Moving from a soil- to ecosystem-centered perspective allows more realistic and simpler hydrological models*" are not only unsubstantiated, but also lack a formalism for parameterization, scale-appropriate governing equations, or tools for systematic hypotheses testing. Certain hydrological processes will always rely on soil properties and microscale physics (Vereecken et al., 2022), while others may manifest "emergent behavior" at larger scales such as catchments and beyond — "emergent" in the sense that they are not predictable from their microscale components such as the complexity of rainfall-runoff based on infiltration theory discussed in Beven (2021). The hydrology community has recognized the significance of ecosystem attributes and hydrologic responses that become apparent at larger spatial scales and over

longer time frames, e.g., from Horton (1933) to the Budyko (1974) framework. Although not explicitly stated, Gao et al. (2023) simply call for "Darwinian hydrology" articulated in Harman and Troch (2014, p.428), ignoring the explicit caveat that "*The Darwinian approach should not be confused with superficially persuasive ad hoc explanations about the holistic interactions that appear to control the regimes of watershed behavior, but do not offer explanations for their origins, or do not provide independent evidence of causation*". The iconic work of Budyko (1974) has been designated "Darwinian" as opposed to "Newtonian" hydrology, as Sposito (2017) explains "because it foregoes reductionist explanations based on constitutive equations in favor of establishing universal relationships based solely on the mass and energy balance laws to which any physical system must conform". The opinion of Gao et al. (2023) with its wholesale rejection of small-scale soil processes offers no such path forward. Most theories and even explanations in textbooks often begin with conceptual hydrologic constructs (perceptual models as defined originally by Beven, 1987) that invoke small scale processes to quantify simple scenarios over uniform soil before embracing the inherent complexity of natural hydrologic systems at larger scales (Koutsoyiannis, 2010), where topography, vegetation, variable climatic patterns jointly lead to hydrologic behavior not anticipated by microscale models of infiltration or runoff (Beven, 2021). Moving from simple to more complex large-scale description of hydrological processes, or even making a conceptual leap to complex ecosystem without ascribing to it untestable traits or intent such as made by Gao et al. (2023) "*According to this view, a terrestrial ecosystem manipulates the soil hydraulic properties to satisfy specific water management strategies*" or "*Our interpretation is that the ecosystems had prepared for this eventuality and had created enough root zone buffer to overcome this period of drought*" would have significantly strengthen the argument for own perspective for large scale hydrology, rather than rejecting the role of present building blocks.

The frustration and debate regarding the role of "reductionist approaches" or small-scale processes in hydrology is not new, as noted by Sivapalan et al. (2003), Harman and Troch (2014), Or (2020) and Beven (2021). We completely agree with Gao et al. (2023) that advances in large scale hydrology in an era of big data and Earth observing platforms require a change in perspective and development of new theories and tools. However, the critique in Gao et al. (2023) that builds a potential shift of perspective on dismissing the significance of soil processes in hydrological studies without offering theoretical alternatives is unwarranted at this stage of development. We envision the emergence of large-scale hydrology characterized by the development of theories and new laws specific to this scale, while acknowledging that small-scale processes continue to influence certain aspects and traits at various levels. Nevertheless, we are grateful for the interest of Gao et al. (2023) in Vereecken et al. (2022) review and the opportunity to address the role of soil processes in contemporary hydrology.

## 3 Soil-centered processes at different scales

Gao et al. (2023) implicitly argue for embracing "Darwinian Hydrology" (Harman and Troch, 2014; Sposito, 2017) as the sole representation of hydrologic processes at large scales while rejecting the role that soil processes might play in this overarching framework. In the absence of a predictive theory for large scale hydrology, ignoring the present understanding of how soil characteristics influence hydrology (including water movement, storage, and availability) at different spatial scales is premature. The limitations of small-scale processes in representing hydrologic behavior in complex natural systems have been examined by many. A recent review by Beven (2021) explains some of the limitations of infiltration theory to describe rainfall-runoff behavior at catchment scales. Hence, we are left with the reality that detailed soil data is crucial at the pedon scale for predicting rainfall partitioning, while for predicting runoff generation at the catchment scale, vegetation and topography become far more important than soil properties. Unlike soil hydrology, catchment hydrology models often do not need to depict details of internal states or process dynamics. In contrast, understanding processes like landslides, groundwater pollution risks from agrochemicals, and subsurface water flow and storage necessitates knowledge of small-scale biological activities (e.g., root growth, microbial, and earthworm activity) affecting hydrological processes. By design, such "subgrid" processes are not captured in large scale or catchment hydrology models. A properly constructed ecosystem-centered modeling framework would significantly reduce details of soil property measurements and embrace landscape traits that dominate at these scales. Such a framework, however, may not adequately address predictions

required for fields or smaller catchments where a detailed representation of processes is needed. Hence, ecosystem-centered and soil-centered approaches are not mutually exclusive but complementary representations of hydrology.

Inquiries continue to concentrate on gaining a process-based comprehension of hydrological variability and its causes across all spatial and temporal scales (Blöschl et al., 2019). This highlights the continuum of scales within which hydrology operates, underscoring that the discipline extends beyond the confines of Gao's perspective. Indeed, environmental issues have underscored the effects on hydrology and the difficulties arising from human influences on the interactions between the water cycle and nature, particularly in complex water management. McDonnell et al. (2021) introduce a "fill-and-spill" concept, explaining how water accumulates in a landscape (fill) until it reaches a critical level (spill), activating new outflow pathways. This process, observed at various scales, suggests that future models should identify the specific scale of interest, highlighting the idea that emergent behaviors depend on the observational scale. Similarly, when combining pedotransfer functions (PTFs) with hydrological models for large-scale modeling, Li et al. (2024) proposes using convolutional neural networks (CNN) as a cross-scale transfer approach. This method reduces potential errors by directly mapping soil and landscape static properties to soil hydraulic parameters across different spatial scales. Intriguingly, certain variables, such as evapotranspiration, allow for upscaling from micro-scale processes like root water uptake, which can be scaled consistently across various levels. Recent developments have led to methodologies for upscaling root water uptake processes and defining effective parameters grounded in micro-scale analyses (e.g. Vanderborght et al., 2023). These methodologies can be easily integrated into both catchment scale and land surface models. A significant advancement in hydrologic modeling is the access to spatially detailed and continuous data, which offers new opportunities for using large-scale system responses to refine parameters, tackle heterogeneity, and enhance model selection and structure. Ideally, the optimal approach involves developing scale-aware parameterization such as multiscale PTFs to span a continuum of scale, considering soil's role in the interconnected geology-plant-atmosphere system vertically such as primary driving concept by water potential gradient (Novick et al., 2022) and horizontally such as hydrological connectivity between different model domains (Janzen et al., 2011).

## 4 Catchment hydrology at a crossroads

Gao et al. (2023) argue for an ecosystem-centered perspective on catchment hydrology (Harman and Troch, 2014) while rejecting the role of small-scale physics that are based first principles ("Newtonian") used to upscale hydrologic processes (e.g. Vereecken et al., 2008). However, they fail to outline a coherent alternative theory for such ecosystem-centered view. McDonnell et al. (2007) have proposed a path forward for building theories suitable for hydrological processes at larger scales; however, not much has been done to translate these concepts into modeling and parameterization tools. Some advances toward a coherent hydrologic theory at catchment scales were made by Reggiani et al. (1999; 1998) for a unified model rooted in thermodynamics with the concept of Representative Elementary Watershed (REW), paralleling the Representative Elementary Volume (REV) concept of soil physics. Reggiani's work meticulously derives conservation laws for mass, momentum, energy, and entropy within a watershed, alongside necessary constitutive relationships and ways for incorporating experimental data and observations into these models. Despite the promise of this modeling approach, Gao et al. (2023) dismiss this effort as "based on the integration of small-scale conservation equations developed for porous media". The point being that, proposing a generic ecosystems' viewpoint without the scientific machinery for generating and testing hypotheses cannot replace fundamental physical and biophysical laws governing hydrologic processes across scales. In this respect, catchment hydrology seems to be at a crossroads with respect to development of its scale-aware scientific basis.

Approaches based on physical principles, applicable at smaller than catchment scales, are crucial in enriching this scientific foundation. With catchment hydrology at a pivotal point in its theoretical development, we believe in integrating ecosystem-based and fundamental physical and biophysical laws derived at smaller scales (Novick et al., 2022). Recent advances in machine learning and deep learning, along with their hydrological applications, may now offer promising avenues to blend physical-based methods (Konapala et al., 2020) and to incorporate soil data across various scales. In contrast, adopting a heuristic "ecosystem scale" approach without scientifically

linked and physically based building blocks harbors the risk of being overwhelmed by advanced machine learning and data driven tools that would render large scale hydrology obsolete.

## 5  Concluding remarks

As Gao et al. (2023) noted, climate models, even the first weather prediction model of Richardson (1922) recognized the importance of mechanistic representation of land surface processes considering water and heat fluxes at the soil surface (Or, 2020). Richardson was not infected by soil-centered bias, and he simply recognized the natural links between soil surface processes and weather models that persist to this day with the inclusion of "small scale" soil processes in global climate models. A 'top-down' approach driven by climate and ecosystem factors may offer certain advantages for catchment hydrology modeling, as shown by Budyko's framework for large catchments and annual balances. However, for certain processes a 'bottom-up' physically based approach remains critical for its explanatory power of localized processes and patterns not resolved by large scale models. A comprehensive large-scale theory would enhance current small-scale foundational elements, adapting to various applications depending on the information and scale of interest. The challenge of explaining catchment-scale behavior's nonlinearities has been transformed into an opportunity for hydrologic model calibration. For instance, the Budyko framework has been applied to examine hydrologic responses to climate change on a continental level (Donohue et al., 2012). Similarly, the complementary relations of Bouchet (1963) exemplify the use of large-scale emergent phenomena to inform evaporation and water balance predictions for extensive areas (Zhang et al., 2010). We observed that the integration of such large-scale emergent behaviors for routine model evaluation milestones has been limited, primarily due to the mismatch in spatial and temporal scales between Earth System Models and hydrologic distributed models (Or, 2020). Despite these challenges, many studies have reported successful applications of these concepts in model evaluation, particularly within the hydroclimatic context.

We believe that both perspectives are needed for hydrology and that a common path should be sought in order to advance hydrology as a discipline in Earth System sciences. The enduring issue of accurately representing small-scale soil processes at the catchment scale is anticipated to be addressed through parameterization using PTFs at an intermediate scale, e.g., 1-km resolution, by incorporating effects of soil structure and vegetation, applying soil-based surface evaporation resistance, and promoting potential synergies among small and large scales with more intimate collaboration between global-scale climate and ecological modelers. Recently, Weber et al. (2024) pointed out the need to further develop hydro-PTFs that go beyond the use of only textural properties. What the viewpoint of Gao et al. (2023) should invoke is the urgent need based on the blueprint of McDonnell et al. (2007) to explore organizing principles that underlie heterogeneity and complexity of catchments instead of attempting to explicitly characterize landscape heterogeneity. Exploring scaling and emergent behaviors, along with network and optimality principles, aligns with a Darwinian approach that aims to understand the origins of these patterns through the processes that generate them (Harman and Troch, 2014). The credibility and applicability of hydrological optimality theory are enhanced when its historical evolution is clarified, guiding its relevance to specific watersheds. Optimality may also explain the self-organization of catchments in Budyko space, where a shape parameter emerges through vegetation's adaptation to climatic conditions in a specific hydrological setting (Hunt et al., 2021; Nijzink and Schymanski, 2022). Indeed, there is a need for increased communication and collaboration between the various disciplines dealing with the hydrology of the land surface across scales to achieve a shared understanding of the challenges and solutions in catchment hydrology. This will help design a consistent and seamless framework for hydrologic research grounded in solid scientific principles.

**Acknowledgments:** We are grateful to Jeff McDonnell and Robert Lee Hill for their valuable feedback on earlier drafts of this article. Ying Zhao is financially supported by the Key Program of the National Natural Science Foundation of China for International Cooperation (42320104006).

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
