# Peer review of "Comment on "Are soils overrated in hydrology?" by Gao et al. (2023)"

_EGUsphere, 2024_

## Referee Comment (RC1)

Comment on "Are soils overrated in hydrology?" by Gao et al. (2023)

Ying Zhao, Mehdi Rahmati, Harry Vereecken, Dani Or

We appreciate the critical commentary of Zhao et al. on our Opinion paper, as it presents an opportunity to clarify our viewpoints. Engaging in such critical discussions is essential for the advancement of the hydrological community and scientific progress as a whole. While we believe that the commentary merits eventual publication, we see the need for substantial improvements to facilitate constructive debate.

1. Clarification of arguments: We note a significant degree of vagueness surrounding the arguments presented in our paper, particularly concerning areas of disagreement. It appears that the authors have extended some of our discussions beyond their original scope, introducing themes that are not directly relevant to our commentary. Examples include the "dichotomy between determinism ('good') and randomness ('evil')" and the distinction between "Darwinian" and "Newtonian" hydrology. We recommend that the authors adhere closely to our arguments, possibly by directly quoting them, to facilitate a more focused and constructive discussion.

2. Addressing misinterpretations: The authors state that "The opinion of Gao et al. (2023) with its wholesale rejection of small-scale soil processes offers no such path forward." If the authors complain about "sweeping and poorly substantiated assertions" in our paper, they should be more precise about our assertions. Nowhere in our paper do we explicitly deny the existence of small-scale processes. Furthermore, the claim that we offer "no path forward" is contradicted by the growing body of research adopting our holistic approach in hydrology, ecology, and land surface modeling (for example, Gao et al., 2014, 2019, 2023; De Boer-Euser et al., 2016; Wang-Erlandsson et al., 2016; Nijzink et al., 2016; Dralle et al., 2020, 2021; Mao and Liu, 2019; McCormick et al., 2020; van Oorschot et al., 2021, 2023; Bouaziz et al., 2022; Hahm et al., 2021, 2024; Liang et al., 2024). We invite the authors to reconsider their interpretation in light of this evidence and clarify how they perceive our stance as lacking a progressive direction.

3. Structural improvements: The commentary currently appears fragmented and lacks a coherent structure. Revising the organization to ensure a logical flow of ideas will enhance readability and comprehension for readers.

4. Incorporation of conceptual diagrams: To better illustrate their arguments, we propose the addition of one or two conceptual diagrams.

Abstract. This comment challenges Gao et al. (2023)'s perspective rejecting the role of soil processes in hydrology.

We do not agree with this summary of our Opinion paper. We never use the word "reject" in our paper. The first sentence of our Opinion paper reads: "Soil is important in hydrology". We declare that this is not about terminology. It is about interpretation. We are afraid that the authors mistook our point of view. The authors can easily see that in our proposed new paradigm (Figure 4b, Gao et al., 2023), soil and soil formation processes are paramount in our proposed ecosystem hierarchy with cause–effect relationships.

We argue that the authors present a false dichotomy between soil-centric and ecosystem-centric views.

Soil-centric and ecosystem-centric views are not a dichotomy, but two different perspectives. The soil-centric view is a reductionist approach, while the ecosystem-centric view is a holistic approach. These perspectives align with the bottom-up and top-down approaches elucidated by Sivapalan et al. (2003), each offering valuable insights depending on the context of the study. We agree that these perspectives can complement each other. However, the authors here are extrapolating our discussion and bringing it to another territory. We think it would be more constructive to stick to the specific assertions of our paper.

These two views of hydrology are complementary and reflect on the inherent multiscale complexity of hydrology where soil processes dominate at certain scales but other processes may become important at catchment scale. We recognize the need for a new scale- aware framework that reconciles the interplay between soil processes at small scales with emergent behaviors driven by vegetation, topography and climate at large scales.

We appreciate the authors' agreement with us that at large scales climate, vegetation, and topography are more important than soil. The authors state that "soil processes dominate at certain scales but other processes may become important at catchment scale". The use of "certain" makes this statement vague. However, assuming that "certain scales" means smaller than catchment scale, then we do not see the disagreement. In our commentary, our focus is on catchment hydrology. Hence, we are interested in determining the dominant controls on catchment scale processes, that is, processes that take a certain minimum area to be operational, which we refer to as the ecosystem scale (e.g. Section 6 in our paper).

We also think that understanding how soil properties are affected by the embodying ecosystem can benefit the description of small-scale processes. Hence, there is a benefit in bringing together the holistic and reductionist approaches. This aspect is recognized in our paper. Even for water quality, solute transport, and transit times studies, the holistic approach is emerging as a promising new framework to represent material transport (Harman and Fei, 2024).

Line 34. "We conclude by suggesting ways to reconcile the two perspectives."

The authors suggest "ways to reconcile the two perspectives". Reconciliation of the two perspectives has been advocated for a long time. Therefore, there is nothing really new in proposing a reconciliation. Moreover, such a reconciliation should be accompanied by a pragmatic way forward, which we feel is lacking in the proposed commentary.

Line 48-52. The assertions in Gao et al. (2023), such as "*the ecosystem, not the soil, determines the land-surface water balance and hydrological processes. Moving from a soil- to ecosystem-centered perspective allows more realistic and simpler hydrological models*" are not only unsubstantiated, but also lack a formalism for parameterization, scale-appropriate governing equations, or tools for systematic hypotheses testing.

In a previous statement, the authors themselves wrote about the "interplay between soil processes at small scales with emergent behaviors driven by vegetation, topography and climate at large scales". Therefore we would appreciate a clarification about where their disagreement starts.

Hydrologists have developed simple conceptual models for a long time that offer "a formalism for parameterization, scale-appropriate governing equations, or tools for systematic hypotheses testing". Moreover, we have referred in our paper to methods to estimate the root zone storage capacity from data, thereby avoiding calibration.

More specifically, in catchment hydrology, many simple hydrological models work very well, e.g. HBV in Europe, Xinanjiang in China, GR4J in France, Sacramento in US. They are not only used for research, but also in water management practice with remarkable robustness, e.g. reservoir operation, runoff prediction, flood mitigation etc.

There is also growing evidence that many parameters of these models can be related to physical attributes. In particular, the parameters of the reservoir representing the root zone compartment of these models can be reliably estimated without soil data. For example, the water retention storage capacity parameter in the Xinanjiang model is the root zone storage capacity, which is evident by over 400 catchments and multi-source and independent landsurface reanalysis data (Liang et al., 2024). The beta parameter is determined by variability in the landscape, and mostly associated with topography (Gao et al., 2017; 2019). The recession parameter Ks is found to be controlled by the characteristics of the aquifer. Interestingly, Ks has almost a constant value for different catchments studies, with diverse climate and landscape (Brutsaert, 2008). All these experiences provide evidence for the co-evolution of climate, landscape, and hydrology.

Line 52.

Again. Please clarify what "Certain hydrological processes" means?

Line 56-58.

Indeed, we did not claim that our idea is novel. However, for a long time the ecosystem-centered perspective was subordinated to soil-centered approach in hydrology. In our commentary, we challenge this hierarchy, aiming to reverse it and elevate the importance of the ecosystem-centered perspective.

Line 67-68. "The opinion of Gao et al. (2023) with its wholesale rejection of small-scale soil processes offers no such path forward."

Answered above.

Line 68-73. Most theories and even explanations in textbooks often begin with conceptual hydrologic constructs (perceptual models as defined originally by Beven, 1987) that invoke small scale processes to quantify simple scenarios over uniform soil before embracing the inherent complexity of natural hydrologic systems at larger scales (Koutsoyiannis, 2010), where topography, vegetation, variable climatic patterns jointly lead to hydrologic behavior not anticipated by microscale models of infiltration or runoff (Beven, 2021).

We agree that textbooks in hydrology are written from a reductionist perspective. However, we also think that this approach can be misleading and that an alternative approach is possible. If we don't move forward, we will continue to use this material to teach our next generations. That is why our Opinion paper is relevant to shift this paradigm. Firstly, many prestigious hydrologists agree that classic textbooks are outdated, including the references provided by the authors (communication with Keith Beven). Secondly, this type of writing is misleading. It gives students a wrong impression that large-scale hydrology study is based on aggregating small-scale knowledge. However, the history of model development shows rather the opposite where many popular models (e.g. HBV, Xinanjiang, GR4J, Sacramento) were developed based on catchment scale data and observations, rather than the small scale understanding. Soils are seldom considered in these models.

Line 81-85.

We thank the authors for endorsing the importance and urgency of our Opinion paper.

Line 104-117.

Firstly, in our opinion paper, we did not wholly reject the necessity of studying small scale properties, especially for water chemistry studies (See Section 6 Limitations): "The variability of soils can have a pronounced influence on predicting water quality, solute transport, and transit times". Secondly, the authors state: "understanding processes like landslides, groundwater pollution risks from agrochemicals, and subsurface water flow and storage necessitates knowledge of small-scale biological activities". This supports the importance of an ecosystem-centered perspective we advocate even in small scale studies.

Line 119.

There are two "across". Please remove one. This sentence is not very clear. Please rephrase.

Line 129-132. "easily integrated into both catchment scale and land surface models".

This is likely an oversell. How can we "easily" integrate it into widely used models? Which catchment scale and land surface models? As catchment hydrologists, we are using HBV, Xinanjiang, GR4J, Topmodel, FLEX etc. How can we use this method in these models?

Line 135-138.

The authors mentioned an "optimal approach". We trust the authors also believe that the hydrological system is not random, and there is a long-time coevolution of climate, geology, and landscape. Co-evolution converges to a (possibly local) optimum, constrained by the Carnot limit (see Kleidon, 2023) of energy conversion. The ecosystem is an integrated system. If we split the system up, and try to optimize merely the elements of the system, it is likely to yield sub-optimal results. The ecosystem optimizes in favor of survival, not in favor of the soil.

Line 145-148.

We have proposed a path forward. Please find our replies above.

Line 148-154.

We did not dismiss the effort of REW. It has been discussed in our opinion paper. Please see Section

2.1.

Line 158-159.

We agree that catchment hydrology is at a crossroads. But we disagree on "development of its scale-aware scientific basis", or in another words "reconcile the two perspectives". Please see our replies above.

Line 163-168.

I cannot follow the logic of these two sentences. The authors argue that machine learning offers promising avenues to incorporate soil data. This is correct. Machine learning can blend and incorporate any data. But then they shift to say "ecosystem scale approach … harbor the risk of being overwhelmed by advanced machine learning". We don't see the logic here. The authors need to elaborate more about their argument. This part needs to be rephrased.

Line 172-177.

The Unreliable intuition on soil hydrology and landsurface process has been discussion in Section 5.2.

Line 193-197.

This sentence is too long and can be better rephrased. The Limitations in the pedotransfer functions approach has been intensively discussed in our Opinion paper Section 2.2. We did not hear the comments on this part from the authors.

Line 197-203.

We have discussed how to "move beyond heterogeneity and process complexity" (McDonnell et al., 2007) in the entire Section 4. In short, we need "putting the terrestrial ecosystem at the centre of hydrology".

References:

Brutsaert, W. (2008), Long-term groundwater storage trends estimated from streamflow records: Climatic perspective, Water Resour. Res., 44, W02409, doi:10.1029/2007WR006518.

Gao, H., Duan, Z., Cai, H (2017). Understand the impacts of landscape features on the shape of storage capacity curve and its influence on flood, Hydrology Research.

Gao, H., Birkel, C., Hrachowitz, M., Tetzlaff, D., Soulsby, C. & H.H.G. Savenije. (2019) A simple topography-driven and calibration-free runoff generation model. Hydrology and Earth System Sciences, 23, 787–809, https://doi.org/10.5194/hess-23-787-2019.

Gao, H., Fenicia, F., and Savenije, H. H. G.: HESS Opinions: Are soils overrated in hydrology?, Hydrology and Earth System Sciences, 27, 2607–2620, https://doi.org/10.5194/hess-27-2607-2023, 2023.

Harman, and Fei (2024). mesas.py v1.0: a flexible Python package for modeling solute transport and transit times using StorAge Selection functions. Geosci. Model Dev., 17, 477–495

Kleidon, A.: Working at the limit: a review of thermodynamics and optimality of the Earth system, Earth Syst. Dynam., 14, 861–896, https://doi.org/10.5194/esd-14-861-2023, 2023.

Liang, J., Gao, H., Fenicia, F., Xi, Q., Wang, Y., and Savenije, H. H. G. (2024) Widespread increase of root zone storage capacity in the United States, EGUsphere [preprint], https://doi.org/10.5194/egusphere-2024-550.

McDonnell, J. J., et al. (2007), Moving beyond heterogeneity and process complexity: A new vision for watershed hydrology, Water Resour. Res., 43, W07301, doi:10.1029/2006WR005467.

Sivapalan, M., Bloschl, G., Zhang, L., and Vertessy, R.: Downward approach to hydrological prediction, Hydrol. Process., 17, 2101–2111, https://doi.org/10.1002/hyp.1425, 2003.

Zhao, Y., Rahmati, M., Vereecken, H., and Or, D.: Comment on "Are soils overrated in hydrology?" by Gao et al. (2023) , EGUsphere [preprint], https://doi.org/10.5194/egusphere-2024-629, 2024.

---

## Author Comment (AC1)

We highly appreciate your review and the suggestions you proposed in response to Gao et al. (2023) 's opinion paper. Taking your suggestions into consideration, we will revise our comments accordingly. For the main points that require discussion or specific responses, we replied and outlined below.

In our revision, we will briefly address the suggestion to make our comment more accessible to audiences like catchment hydrologists. In respect to the comments related to scaling issues such as REW applicability and emergent properties, we will clarify the content provided in the lines 201-204 "*Exploring scaling and emergent behaviors, along with network and optimality principles, aligns with a Darwinian approach that aims to understand the origins of these patterns through the processes that generate them (Harman and Troch, 2014). The credibility and applicability of hydrological optimality theory are enhanced when its historical evolution is clarified, guiding its relevance to specific watersheds.*" and lines 125-129 "*This process, observed at various scales, suggests that future studies should identify the specific scale of interest, highlighting the idea that emergent behaviors depend on the observational scale. Similarly, certain variables, such as evapotranspiration, allow for upscaling from micro-scale processes like root water uptake, which can be scaled consistently across various levels.*"

Concerning the development of PTFs, we will incorporate the new suggestions by citing Weber et al. (2024). Additionally, we will reference another recent paper by Li et al. (2024), presented at the International Soil Modelling Consortium (ISMC) in Tianjin, which proposes using convolutional neural networks (CNN) as a cross-scale transfer approach. In line with your considerations, we will also emphasize the primary driving concept for fluxes, as highlighted by Novick et al. (2022), to address the advancements in soil-biophysical and hydropedological theory.

We appreciate the reviewer's three suggestions to substantiate our points regarding the role of soils: i) work out the essential biophysical controls by soils, ii) highlight the theoretical advances and how they could be incorporated in simplified hydrological models and iii) elaborate the theoretical necessity of biophysical principles more clearly. While we agree with points i) and iii), we consider addressing these suggestions in more detail beyond the scope of our comment. For suggestion ii), we will further clarify the statements made in lines 129-138 "*Recent developments have led to methodologies for upscaling root water uptake processes and defining effective parameters grounded in micro-scale analyses (e.g. Vanderborght et al., 2023). These methodologies can be easily integrated into both catchment scale and land surface models. A significant advancement in hydrologic modeling is the access to spatially detailed and continuous data, which offers new opportunities for using large-scale system responses to refine parameters, tackle heterogeneity, and enhance model selection and structure. Ideally, the optimal approach involves developing scale-aware parameterization for such models such as multiscale PTFs to span a continuum of scale, considering soil's role in the interconnected geology-plant-atmosphere system such as hydrological connectivity between different model domains (Janzen et al., 2011).*"
*Finally we see the need for a joint community effort to move forward some of the more fundamental issues raised by this reviewer and reviewer 3.*

**References:**
Harman, C. and Troch, P.: What makes Darwinian hydrology" Darwinian"? Asking a different kind of question about landscapes, Hydrology and Earth System Sciences, 18, 417-433, 2014.
Janzen, H., Fixen, P., Franzluebbers, A., Hattey, J., Izaurralde, R. C., Ketterings, Q., Lobb, D., and Schlesinger, W.: Global prospects rooted in soil science, Soil Science Society of America Journal, 75, 1-8, 2011.
Li, P., Zha, Y., Zhang, Y., Tso, C.-H. M., Attinger, S., Samaniego, L., & Peng, J. Deep learning integrating scale conversion and pedo-transfer function to avoid potential errors in cross-scale transfer. Water Resources Research, 60, e2023WR035543. 2024.

Novick, K. A., Ficklin, D. L., Baldocchi, D., Davis, K. J., Ghezzehei, T. A., Konings, A. G., MacBean, N., Raoult, N., Scott, R. L., Shi, Y., Sulman, B. N., and Wood, J. D.: Confronting the water potential information gap, Nat Geosci, 15, 158–164, https://doi.org/10.1038/s41561-022-00909-2, 2022.

Vanderborght, J., Leitner, D., Schnepf, A., Couvreur, V., Vereecken, H., and Javaux, M.: Combining root and soil hydraulics in macroscopic representations of root water uptake, Vadose Zone Journal, 2023. e20273, 2023.

Weber, T. K. D., Weihermüller, L., Nemes, A., Bechtold, M., Degré, A., Diamantopoulos, E., Fatichi, S., Filipović, V., Gupta, S., Hohenbrink, T. L., Hirmas, D. R., Jackisch, C., de Jong van Lier, Q., Koestel, J., Lehmann, P., Marthews, T. R., Minasny, B., Pagel, H., van der Ploeg, M., Svane, S. F., Szabó, B., Vereecken, H., Verhoef, A., Young, M., Zeng, Y., Zhang, Y., and Bonetti, S.: Hydro-pedotransfer functions: A roadmap for future development, EGUsphere [preprint], https://doi.org/10.5194/egusphere-2023-1860, 2023.

---

## Author Response (AR1)

**Editor Report:**

Public justification (visible to the public if the article is accepted and published):

Dear Dr. Zhao,

I very much enjoyed reading your scientific comment, the reviewer comments and your related replies. When accepting the OP paper of Gao et al. (2023) for review in 2023, I was convinced that it would stimulate an interesting and necessary debate about a key problem that "is well known, but not well done" in hydrology. The quality of your comment and the related discussion, shows that the debate has extended even beyond the discussion phase of the OP. HESS is definitely the best forum such a debate and I am more than happy to host both the OP paper of Gao et al. (2023) and your scientific comment in HESS.

Being a "believer" in soil physics myself, I absolutely agree that soils play a crucial role in the hydrological cycle. There is no runoff without soils, no soil moisture storage without capillarity, neither roots can grow nor rivers can be formed without soils.

Yet I think that the OP of Gao et al. (2023) reflects, in a provocative way, that your community has been still split about the question "how catchments work" and how to make use of soil physical knowledge in this context for a long time.

During my PhD, which was on reactive transport, I "discovered" that the soil is engineered by earthworms. Worm burrows have a crucial impact on solute transport and surface runoff generation (no news), yet these influences are not straightforward to capture with soil hydraulic functions. Vegetation mediates via plant transpiration the largest hydrological flux on Earth, which means that vegetation dynamics and ecosystem adaption to it's (changing) niche, are key drivers for hydrological change. Yet, I personally think that a "state based" model paradigms are not appropriate to capture such adaptations. Ecological optimality is a testable and promising hypothesis in this respect, as reflected in the recent work of Nijzink et al (2022) and Hunt et al (2021)

That said I am very much inclined to follow the recommendations of Siva Sivapalan and Reviewer 4 i.e. to accept your comment as it stands, and of course I agree that we need more constructive cooperation across both paradigms to move forward and hopefully cut the Gordian knot.

Yet I grant you the possibility for final adjustments of your manuscript in line with your replies.

Looking forward to receive the revised manuscript,

Erwin Zehe

References:

Nijzink, R. C. and Schymanski, S. J.: Vegetation optimality explains the convergence of catchments on the Budyko curve, Hydrol. Earth Syst. Sci., 26, 6289–6309, https://doi.org/10.5194/hess-26-6289-2022, 2022.

Hunt, A. G., Faybishenko, B., and Ghanbarian, B.: Predicting Characteristics of the Water Cycle From Scaling Relationships, Water Resour. Res., 57, e2021WR030808, https://doi.org/10.1029/2021WR030808, 2021

**Reply:**

Dear Editor Dr. Zehe,

We greatly appreciate your evaluation and the information you shared. This not only helps establish a consistent framework for hydrologic research based on solid scientific principles but also fosters lively discussions and a collaborative research atmosphere, which is particularly important for the OP paper. As mentioned, we look forward to organizing or joining a workshop to develop a shared understanding of the challenges and solutions in catchment hydrology.

We have revised our manuscript by incorporating your suggestions and our responses to the reviewer's comments. If you have any further requirements, please let us know.

Regards

Ying Zhao